# Fat-Corrected Pancreatic *R2\** Relaxometry from Multi-Echo Gradient-Recalled Echo Sequence Using Convolutional Neural Network

Maria Filomena Santarelli [1], Sara Joubbi [1], Antonella Meloni [2,3], Laura Pistoia [2], Tommaso Casini [4], Francesco Massei [5], Pier Paolo Bitti [6], Massimo Allò [7], Filippo Cademartiri [2] and Vincenzo Positano [2,3,*]

1 CNR Institute of Clinical Physiology, 56124 Pisa, Italy
2 Department of Radiology, Fondazione G. Monasterio CNR-Regione Toscana, 56124 Pisa, Italy
3 U.O.C. Bioingegneria, Fondazione G. Monasterio CNR-Regione Toscana, 56124 Pisa, Italy
4 Centro Talassemie ed Emoglobinopatie, Ospedale "Meyer", 50139 Firenze, Italy
5 UO Oncoematologia Pediatrica, AOUP Santa Chiara, 56124 Pisa, Italy
6 Servizio Immunoematologia e Medicina Trasfusionale—Dipartimento dei Servizi, Presidio Ospedaliero "San Francesco" ASL Nuoro, 08100 Nuoro, Italy
7 Ematologia Microcitemia, Ospedale San Giovanni di Dio—ASP Crotone, 88900 Crotone, Italy
* Correspondence: positano@ftgm.it; Tel.: +39-0503152613

**Abstract:** Fat-corrected *R2\** relaxometry from multi-echo gradient-recalled echo sequences (mGRE) could represent an efficient approach for iron overload evaluation, but its use is limited by computational constraints. A new method for the fast generation of *R2\** and fat fractions (FF) maps from mGRE using a convolutional neural network (U-Net) and deep learning (DL) is presented. A U-Net for the calculation of pancreatic *R2\** and FF maps was trained with 576 mGRE abdominal images and compared to conventional fat-corrected relaxometry. The U-Net was effectively trained and provided *R2\** and FF maps visually comparable to conventional methods. Predicted pancreatic *R2\** and FF values were well correlated with the conventional model. Estimated and ground truth mean *R2\** values were not significantly different ($43.65 \pm 21.89$ vs. $43.77 \pm 19.81$ ms, $p = 0.692$, intraclass correlation coefficient-ICC = 0.9938, coefficient of variation-CoV = 5.3%), while estimated FF values were slightly higher in respect to ground truth values ($27.8 \pm 16.87$ vs. $25.67 \pm 15.43$ %, $p < 0.0001$, ICC = 0.986, CoV = 10.1%). Deep learning utilizing the U-Net is a feasible method for pancreatic MR fat-corrected relaxometry. A trained U-Net can be efficiently used for MR fat-corrected relaxometry, providing results comparable to conventional model-based methods.

**Keywords:** convolutional neural network; fat-corrected relaxometry; pancreas; iron overload; U-Net

## 1. Introduction

With the consequent damage to several organs such as the liver, heart, and pancreas, iron overload represents a critical health problem in patients with primary and secondary hemochromatosis [1,2]. The most important pathology associated with secondary hemochromatosis is beta-thalassemia major, the most common genetic disorder worldwide [3]. Thalassemia major is a multi-organ disease; hence the proper management should include iron overload quantification in the liver, heart, pancreas, and other organs [4].

Magnetic resonance imaging (MRI) represents the most established approach for iron overload evaluation, as iron deposits increase the magnetic field heterogeneities, resulting in increased R2 and *R2\** relaxation values that lead to a decline in MRI signal proportionally to the importance of iron overload. Among several proposed techniques, *R2\** relaxometry by the use of multiecho gradient-echo (mGRE) sequences is the most established approach with an important number of applications, including iron quantification in the liver, heart [5], pancreas [6], kidney [7], and brain. In *R2\** relaxometry, *R2\** values are estimated by fitting the signal intensity at each pixel of the images acquired at multiple echo times (TEs) with an

appropriate decay model [8]. The result of the fitting method is an *R*2* map with the same dimensions of the originating images. The setting of the signal fitting process should take into account several acquisition/physical aspects that deviate the measured signal from the theoretical one [9], such as the limited range of practicable TEs (about 1–20 ms), the presence of MR rectified noise [10], and the chemical shift effect induced by the presence of fat [11]. The latter issue is particularly challenging if the organ under examination is subjected to fat infiltration, as it usually happens in the liver and pancreas. Hence, the development of methods for the compensation of the chemical shift effect in *R*2* measurement is important in the diagnosis of iron overload. Fat modulates the signal decay generating a periodic signal fluctuation, with a period of about 4.52 ms at 1.5 T, over-imposed to the exponential decay. To minimize the effect of fat on *R*2* measurement, the out-of-phase/in-phase GRE sequence (using a 2.26 ms TE interval, half of the oscillation period) is largely adopted in the clinical setting for *R*2* quantification [12,13]. In this sequence, the signal oscillation is "symmetric" with respect to the pure exponential decay and can be compensated by the fitting algorithm. However, the accuracy of this approach is invalidated by the interference of multiple spectral peaks of fat [14,15]. Fat suppression (FS) techniques could be used to minimize the fat signal [16]. However, FS methods increase the acquisition time and are less efficient in the presence of iron overload. In pancreatic iron overload, the progressive manual removal of TEs corresponding to the strongest deviations from the theoretical exponential decay was demonstrated to be reproducible and led to results comparable with measurements on FS images [17]. The best approach is represented by the use of fitting models that separate the multi-peak fat signal from the water contribution (fat-corrected relaxometry), firstly proposed for *R*2* quantification in the liver [15,18] and later extended to the pancreas [19,20]. A further advantage of this approach is to make available an additional clinical value of the MRI exam by estimating the fat fraction (FF) in the pancreas. However, the fat-corrected relaxometry process is generally very slow due to the use of multi-start iterative optimization algorithms, which are necessary to obtain a correct convergence of the pixel-wise fitting procedure [19,21,22]. Hence, the development of near real-time algorithms for the generation of *R*2*/FF maps could favourite the spread of this technique in the clinical setting, allowing both a more precise *R*2* measure and evaluation of fat deposition in thalassemia patients. In fact, the integration of *R*2*/FF maps generation in clinical software requires near real-time image processing, as the software is used in an interactive manner to explore different pancreatic regions.

Recently, Convolutional Neural Networks (CNN) have become extremely popular, especially in medical image processing. One widely used CNN is the so-called U-net, specifically designed for image segmentation [23] and also applicable to image-to-image translation tasks [24]. Several studies exploited the U-Net design to solve the water/fat separation problem from mGRE sequences [25–28], while the problem of fat-corrected relaxometry is less explored. In particular, Andersson et al. [25] performed whole-body FF mapping from 5-TEs sequences, without computation of *R*2* maps. Cho and Park [26] performed water-fat separation from 6-TEs sequences, obtaining water and fat images as the output of the U-Net network. A similar approach was proposed by Liu et al. [27] by using a MEBCRN network, including a feature extraction module and a water–fat separation module. The input of the MEBCRN network was represented by real and imaginary 8-TEs sequences. Finally, Goldfarb et al. [28] used a cardiac dark blood real/imaginary 12-TEs sequence as input of a U-Net network, while the output was represented by four channels (water, fat, *R*2*, and off-resonance maps).

As both *R*2* and FF mapping are needed for the iron overload assessment in the pancreas, an effective U-Net design should provide both FF and *R*2* maps as output, as in the Goldfarb approach [28]. Moreover, as in the clinical setting, where only magnitude images are provided by the scanner, the input of the U-Net should be represented by magnitude image sequences instead of real/imaginary images. Finally, the computational efficiency of the trained U-Net should be assured to overcome the limits of the standard curve-fitting approach, as previously described.

The objective of the present study is to develop a computationally efficient method, based on a deep learning approach, to obtain the precise and fast generation of fat-corrected pancreatic *R2\** and FF maps from mGRE magnitude images acquired to estimate the iron overload level in the pancreas.

## 2. Materials and Methods

### 2.1. Fat-Corrected Relaxometry

*R2\** relaxometry allows the quantitative evaluation of the *R2\** parameter in a tissue based on mGRE MR sequences. *R2\** relaxometry requires the acquisition of a series of images at increasing TEs, that can be performed after a single excitation pulse in mGRE sequences. *R2\** values are estimated by fitting the signal intensity from the tissue of interest at multiple TEs with an appropriate decay model. To compensate for the interference of multiple spectral peaks of the fat that deviate the measured signal from the theoretical exponential decay, the effect of fat should be introduced in the decay model. Accordingly, for magnitude MR images that are commonly used in clinical practice, the signal decay can be modeled as [15]:

$$S\left(TE, \rho_w, \rho_f, R2^*\right) = \left| \rho_w + \rho_f \left( \sum_{n=1}^{N} \alpha_n e^{j2\pi f_n TE} \right) e^{-R2^* TE} \right|, \tag{1}$$

where $\rho_w$ and $\rho_f$ are the amplitudes of water and fat signals, and *TE* is the echo time. The *R2\** values of water and fat ($R2_w^*$ and $R2_f^*$, respectively) are usually modeled as a unique value *R2\**, as described in Equation (1). *fn* are the frequencies for the fat spectral peaks and $\alpha_n$ are the relative amplitudes of the fat signal, such that $\sum_{n=1}^{N} \alpha_n = 1$. Previous studies suggested that N = 6 provides a correct estimation of the fat spectrum. The fat spectrum depends on the nature of the fat tissue, as demonstrated in spectroscopy studies [29,30], and their parameters can be fixed once the organ under study has been defined. Hence, the signal model described in Equation (1) has three unknown parameters to be estimated: $\rho_w, \rho_f,$ and $R2^*$. The fat-corrected magnitude fitting procedure herein described can be implemented by a descent-based nonlinear least-squares (NLLS) fitting, that also compensates for B0 field inhomogeneities [31], where all unknown parameters ($\rho_w$, $\rho_f$, *R2\**, and B0 field) are estimated jointly [31,32]. As NLLS is a local optimizer, the estimated parameters' values will depend on the initial values and the limits of the search space.

Fat-corrected relaxometry is a computationally expensive procedure, as the fitting procedure could be easily trapped in local minima due to the low number of available signal samples (6–10 in most applications). Hence, computationally expensive global optimization approaches such as multi-start are often needed [19]. When the procedure is iterated to obtain pixel-wise *R2\** and FF maps, the required time could become not compatible with the clinical practice.

### 2.2. Ground Truth

Images from 192 thalassemia major patients (92 males and 100 females, age 7–58 years, mean age 39.5 ± 16.2 years) were retrospectively studied. All patients were consecutively enrolled from years 2009 to 2020 in the core lab of the MIOT/eMIOT (Myocardial Iron Overload in Thalassemia) network, constituted by thalassemia and MRI centers where MRI exams are performed using homogeneous, standardized, and validated procedures and where patients' clinical–instrumental data are collected in a centralized, web-based database [33,34]. The study complied with the Declaration of Helsinki. All subjects or their parents gave written informed consent to the protocol. The project was approved by the institutional ethics committee.

For each patient, a specific MR data set acquired for assessing the iron overload level in the pancreas was used in the study. Each data set included ten to twelve axial slices covering the abdomen, including the liver and pancreas, obtained by an *R2\** mGRE sequence [6]. Each slice (thickness 8.0 mm) was acquired at ten echo times (first TE 2.0 ms, echo spacing

of 2.26 ms) in a single end-expiratory breath-hold. The echo spacing was chosen equal to the separation between the in-phase and out-phase conditions of the fat/water interface to minimize the shift of signal decay. An MRI was performed using a 1.5 T MRI scanner (Signa Excite HD or Signa Artist, GE Healthcare, Milwaukee, WI) with a cardiac phased-array receiver surface coil (flip angle 25°, matrix 192 × 192 pixels, field of view (FOV) 40 × 40 cm, bandwidth 62.5 KHz, number of excitations 1).

Images from each patient (Figure 1a) were associated with the corresponding clinical labels, as defined in the clinical assessment of patients [6]. Labels were defined using a custom-written, previously validated software (HIPPO-MIOT® v2.0, FTGM, Pisa, Italy). Three regions of interest (ROIs) were manually drawn over the head, body, and tail of the pancreas, respectively, encompassing the parenchymal tissue and taking care to avoid confounding anatomy (e.g., large blood vessels or ducts) and areas involved in susceptibility artifacts from gastric or colic intraluminal gas. ROIs stored in the HIPPO-MIOT software database were exported as binary masks (Figure 1b). For each ROI mask, a squared region (64 × 64) centered on the mask was defined and propagated on all the images at different TEs, obtaining a collection of 576 (192 × 3) multi-echo MR images (of size 64 × 64 × 10) (Figure 1c). Each multi-echo image was processed by the fat-compensated relaxometry procedure previously described, using a validated code [15,31] (Figure 1d) to obtain 576 pairs of 64 × 64 *R*2* and FF maps (Figure 1e). The initial conditions of the NNLS optimizer were tuned based on the expected range of *R*2* and FF values in the pancreas. Hence, the knowledge base for the deep learning algorithm was constituted by 576 mGRE 64 × 64 × 10 images and 576 corresponding pairs of 64 × 64 *R*2* and FF maps.

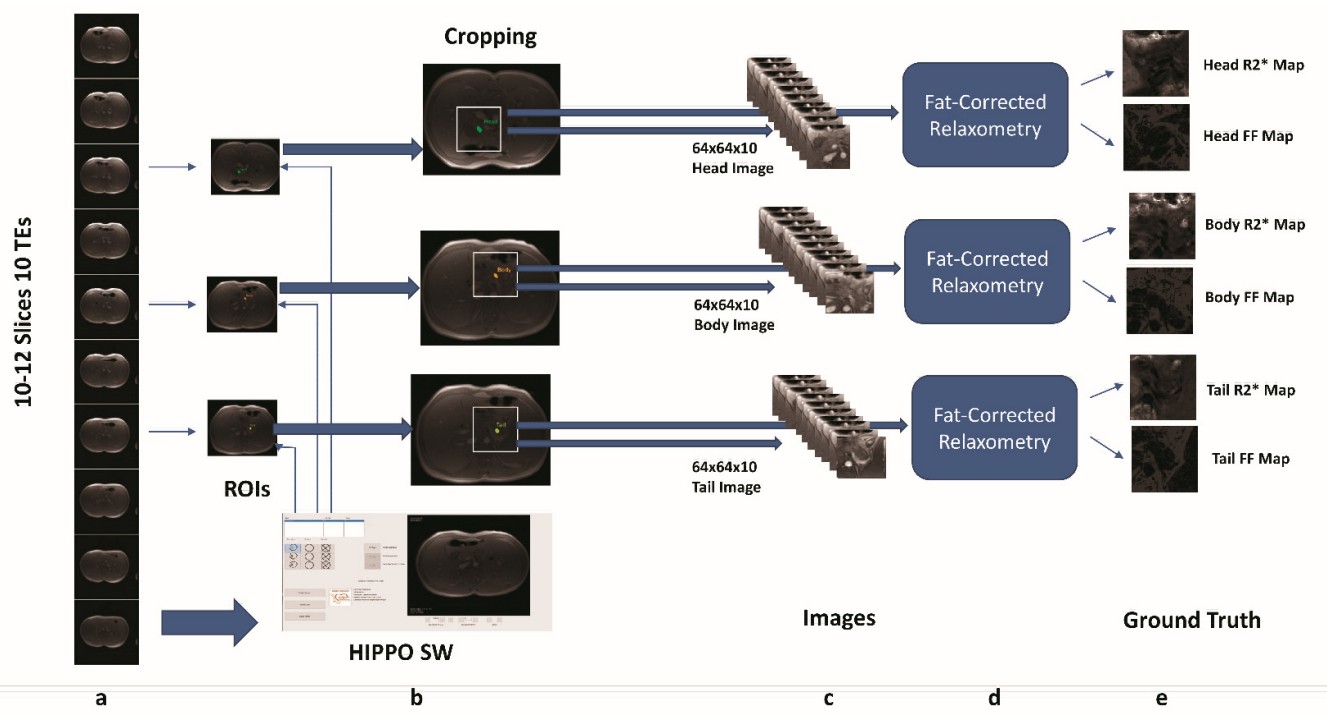

**Figure 1.** Ground truth data generation. (**a**) original MR image data; (**b**) ROIs encompassing pancreatic regions; (**c**) cropped 64 × 64 × 10 MR images; (**d**) Fat-corrected relaxometry processing; (**e**) corresponding *R*2* and FF maps representing the Ground Truth for U-Net training and validation.

*R*2* and FF values corresponding to the three pancreatic regions were obtained by computing the median value of *R*2* and FF of the pixels within the head, body, and tail ROIs defined in the clinical assessment of patients.

### 2.3. U-Net Model

The U-Net network was developed based on the model of Goldfarb el al [28]. The input to the network was represented by mGRE images (10 frames with 64 × 64 pixels), obtained from the original MR images by the procedure previously described. The U-Net architecture was selected to generate relaxometry and FF maps [23,24]. Hence, a 10-input 2-output channel 2D U-Net (one channel for each of the 10 TEs in input, two channels for $R2*$ and FF maps as output) was used (Figure 2). The encoder has three layers, with 96, 192, and 284 filters. Each layer of the encoder block includes a 2D convolution filter, a batch normalization, and a ReLU activation, followed by a max-pooling filter. The bottleneck layer is constituted by 768 2D convolution filters. The decoder blocks include an up-sampling filter, a 2D convolution filter, a batch normalization, and a ReLU activation. A 3 × 3 kernel size was used in all convolution filters. The last layer is a 2D convolutional layer with linear activation. The network included a total of 8,534,114 trainable parameters.

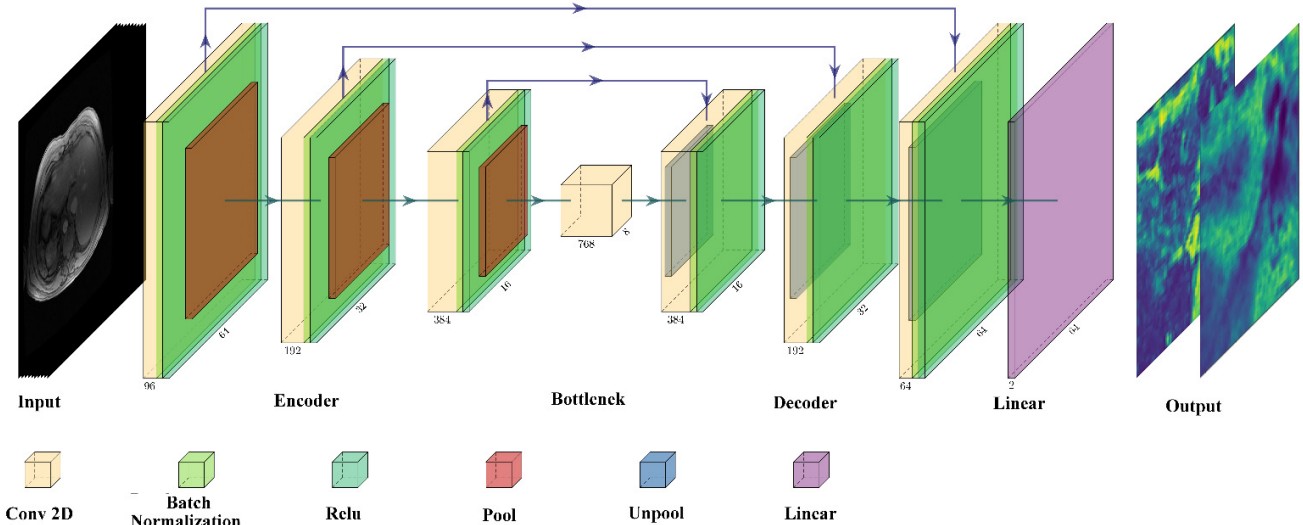

**Figure 2.** U-Net architecture.

The loss function was defined as the sum of mean squared errors (MSE) between the U-Net's $R2*$ and FF maps and the ones generated by the fat-compensated relaxometry (ground truth). An L2 kernel regularization of 0.2 was applied to each 2D convolution layer. The accuracy metric was defined as the mean of the normalized Root Mean Square Error (RMSE) between generated and ground truth $R2*$ and FF maps.

The similarity between ground truth $R2*$ and FF maps and U-Net output was assessed by the normalized root MSE (NRMSE) and Structural Similarity Index (SSI) values. The NRMSE is defined as:

$$\text{NRMSE}(G, U) = \sqrt{\frac{1}{n}\sum_i |G_i - U_i|^2}, \tag{2}$$

where $n$ represents the number of map pixels, $G_i$ is the gray value of a single pixel of the ground truth map and $U_i$ is the gray value of a pixel from the U-Net map. NRMSE represents the mean estimation error in a map pixel. The SSI index is expressed as [35]:

$$\text{SSI}(G, U) = [l(G, U)]^\alpha \cdot [c(G, U)]^\beta \cdot [s(G, U)]^\gamma, \tag{3}$$

where $l(G, U)$ is the luminance comparison function, $c(G, U)$ is contrast comparison function, $s(G, U)$ is the structure comparison function and $\alpha > 0$, $\beta > 0$, $\gamma > 0$ denote the relative importance of each of the metrics. We assume $\alpha = \beta = \gamma = 1$ in the present study. A value of SSI = 1 indicates the maximum similarity between the two maps.

### 2.4. U-Net Implementation, Training, and Testing

The U-Net model was implemented in the Tensorflow 2.5/Keras environment. Network training was performed on an Intel Core i7 5.1 MHz PC, 32 Gb RAM, equipped with an NVIDIA RTX3090 GPU with 24 Gb RAM. Of the whole dataset of mGRE images, 80% were used for network training (461), and the remaining 20% of images (115) were used for the final test. Iterative train and validation were carried out performing each validation at the end of each epoch, each epoch including 20 steps. The validation set size was set to 20% of the training data set. Nadam optimizer (initial learning rate $10^{-4}$, β1 = 0.9, β2 = 0.999) was used with a batch size of 8. Early stopping of the training process after 200 epochs with no increment of validation accuracy metric was adopted. Cross-validation was applied, consisting of performing the training five times with a different distribution of cases among the training and validation sets. Best training and validation accuracy metric values for each training process were recorded.

### 2.5. Statistical Analysis

Data were analyzed using MedCalc version 20.015 (MedCalc Software Ltd., Ostend, Belgium) statistical package. Continuous variables were described as mean ± standard deviation (SD). For continuous values with normal distribution, comparisons between groups were made by a paired *t*-test (for two groups) or a one-way ANOVA (for more than two groups). Normality was assessed by the Kolmogorov–Smirnov test. CoV was obtained as the ratio of the SD of the half mean square of the differences between the repeated values, to the general mean. The intraclass correlation coefficient (ICC) was obtained from a two-way random effects model with measures of absolute agreement. The Bland–Altman (BA) technique was used to plot the absolute difference versus the ground truth values. Bias was the mean of the difference between the two methods, and agreement was the mean ±1.96 SDs. In all tests, a 2-tailed probability value of 0.05 was considered statistically significant.

## 3. Results

### 3.1. Ground Truth

The mean $R2^*$ values in the ground truth data were 44.80 ± 24.71 s$^{-1}$ in the head, 43.52 ± 23.79 s$^{-1}$ in the body, and 43.77 ± 24.03 s$^{-1}$ in the tail. The mean FF values were 26.35 ± 15.37% in the head, 26.70 ± 16.91% in the body, and 27.85 ± 17.09% in the tail, respectively. No significant difference was found in $R2^*$ and FF data between the three pancreatic regions. $R2^*$ values ranged from 6 to 136 s$^{-1}$, covering the range from absent to severe iron overload.

The mean processing time for the generation of $R2^*$/FF maps was 490 ± 140 ms.

### 3.2. U-Net Results

In the network training and testing, the assessed accuracy was 5.02 ± 0.26 and 5.69 ± 0.99 for the validation and test set, respectively. The mean number of iterations needed to reach convergency was 1679 ± 321.

The mean NRMSE values for the test set were 5.41 ± 0.64 s$^{-1}$ and 5.12 ± 0.47% for the $R2^*$ and FF maps, respectively ($p = 0.4544$). The mean SSI values were 0.95 ± 0.01 and 0.88 ± 0.02 for $R2^*$ and FF maps, respectively. A significant difference ($p = 0.0034$) was found between SSI values for $R2^*$ and FF maps. No significant difference was found in NRMSE and SSI values among the three pancreatic regions. Figure 3 compares the ground truth and the estimated $R2^*$ and FF maps for a patient in the test set (body, patient #40). The estimation error was low in both $R2^*$ and FF maps in the clinical target.

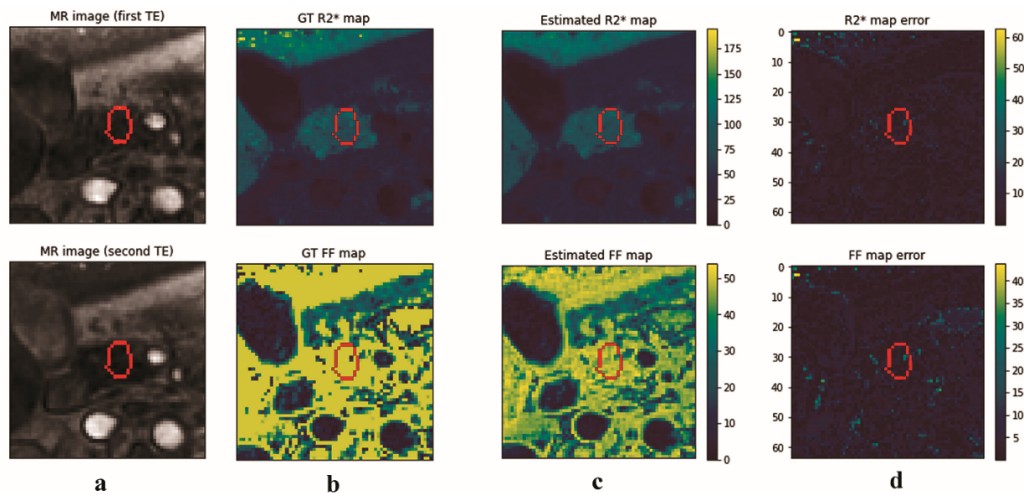

**Figure 3.** Comparison of ground truth (GT) and estimated $R2^*$ and FF maps. The ROI traced for the clinical measurement is over-imposed in red. (**a**) MR image (first and second TEs displayed); (**b**) GT $R2^*$ and FF maps; (**c**) estimated $R2^*$ and FF maps; (**d**) absolute difference between GT and estimated maps.

Limiting the analysis to the pancreatic region, the mean NRMSE values were $3.75 \pm 0.38$ s$^{-1}$ and $4.07 \pm 0.62\%$ for the $R2^*$ and FF maps, respectively ($p = 0.3832$).

Estimated and ground truth mean $R2^*$ values evaluated in the three pancreatic regions were not significantly different ($R2^*$: $43.65 \pm 21.89$ vs. $43.77 \pm 19.81$, $p = 0.692$, ICC = 0.9938, CoV = 5.3%). A significant difference was found in FF values ($27.8 \pm 16.87$ vs. $25.67 \pm 15.43$, $p < 0.0001$, ICC = 0.986, CoV = 10.1%). No significant difference was found in estimated $R2^*$ and FF values among the three pancreatic regions (Figure 4). Figure 5 shows the Bland–Altmann plots comparing estimated and ground truth $R2^*$ (Figure 5a) and FF values (Figure 5b). A good correspondence was found between ground truth and estimated $R2^*$ values, with a negligible bias ($-0.12$) and a 95% confidence interval for the difference of $[-6.55:6.31]$. The developed U-Net slightly underestimated the FF value ($-2.13\%$) with a 95% confidence interval $[-4.08:8.35]$. The mean processing time for the generation of $R2^*$/FF maps by the trained U-Net was $21.9 \pm 2.46$ ms.

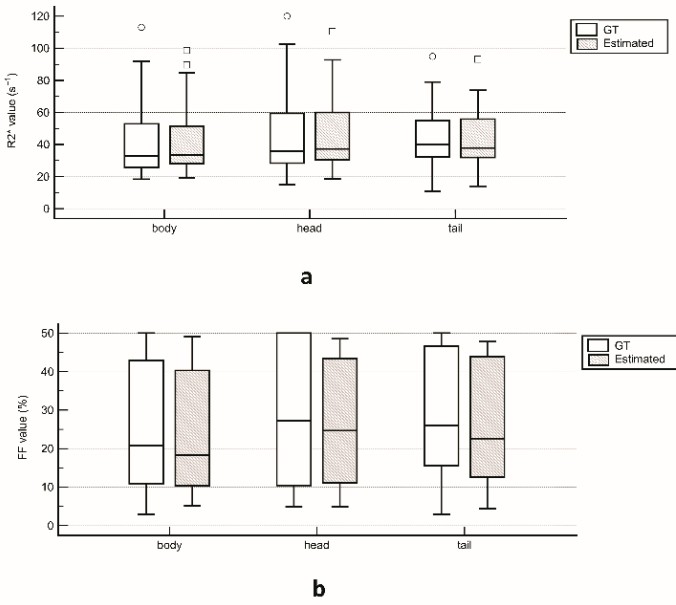

**Figure 4.** Ground Truth (GT) and estimated $R2^*$ (**a**) and FF (**b**) values among the three pancreatic regions.

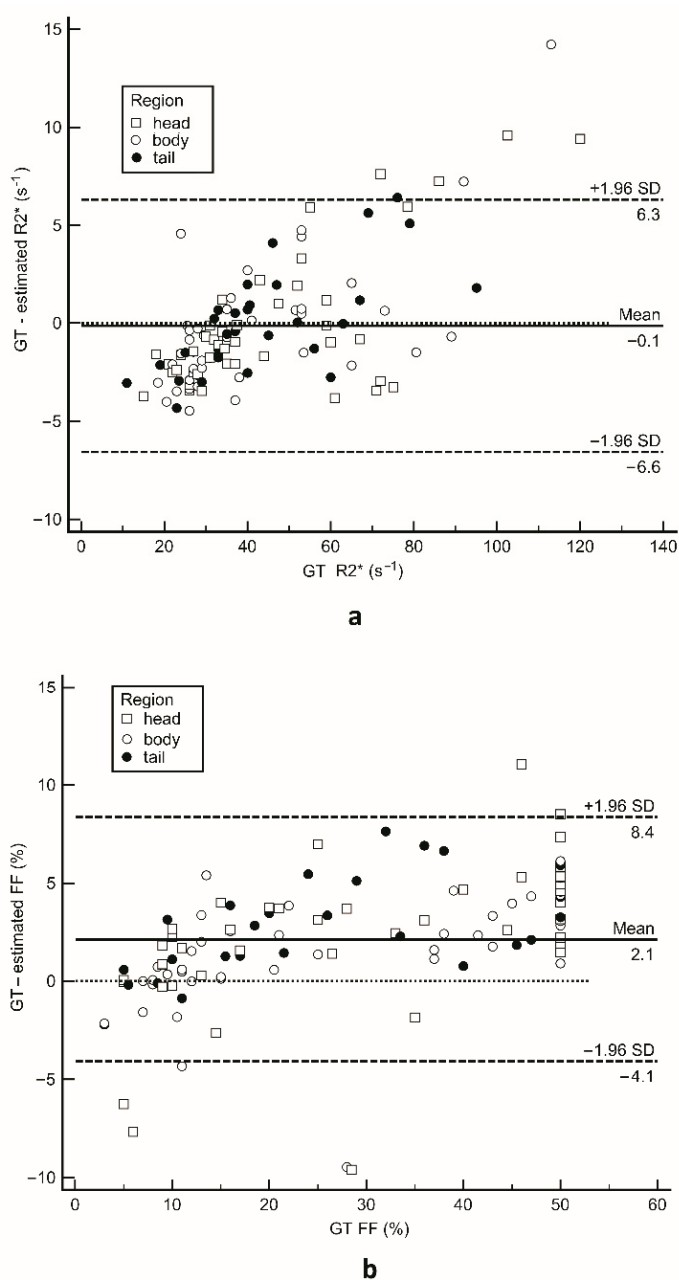

**Figure 5.** Bland–Altman plots reporting the difference between Ground Truth and estimated *R*2* (**a**) and FF (**b**) values among the three pancreatic regions.

## 4. Discussion

A robust and noninvasive evaluation of pancreatic iron by MRI is strongly desirable to prevent diabetes and preserve pancreatic reserve [36,37]. Moreover, several MRI studies demonstrated that pancreatic iron overload is a prospective marker of cardiac iron risk [38,39]. The accurate estimate of pancreatic iron overload may be confounded by the fat infiltration, causing an incorrect estimate of the *R*2* value. Hence, the spread of fat-corrected relaxometry in clinical practice is desirable. Deep learning represents a promising technique to improve MR relaxometry in terms of speed, efficiency, and quality, although this field is less explored than others [40]. Several studies proposed using a deep learning approach for T1 [41] and T2 [42] relaxometry, but in our knowledge, there are no studies about the use of CNN for fat-correction relaxometry. U-Net architectures similar to the one presented in our study were proposed to address the water/fat separation problem, with different anatomical targets [25–28]. Our approach is similar to the one proposed

by Goldfarb et al. [28], that used a cardiac dark blood real/imaginary 12-TEs sequence as the input of a U-Net network, while the output was represented by four channels (water, fat, $R2^*$, and off-resonance maps). In respect to the Goldfarb solution that employed a classic U-Net architecture with four encoded layers, we used three encoder layers to compensate for the smaller input image size. Differently from the Goldfarb approach, our U-Net network processed magnitude mGRE images, as usually acquired in a clinical practice in iron overload studies. This approach could represent an advantage as an FF dedicated MR sequence is not needed and retrospective studies could be conducted on magnitude images acquired for $R2^*$-only measurements. Moreover, the pancreas represents a complex structure affected by a high chemical shift effect due to infiltrating fat. The FF value reached 30% in thalassemia patients [20]. Hence, the population involved in the present study represents a challenging test for the development of an efficient FF/$R2^*$ map generation method.

Ground truth data collected in the study represents a typical well-treated thalassemia major patient population, with a mean $R2^*$ value of about 44 s$^{-1}$ (T2$^*$ = 22 ms), slightly lower in comparison with the lower limit of the normal T2$^*$ value (26 ms) [43]. As expected from previous studies, no significant difference in the $R2^*$ value was found among the tree pancreatic regions [17,20,43,44]. The FF value was about 26%, confirming the higher fat content in thalassemia patients with respect to the normal population [20,44].

The developed network was able to generate $R2^*$ and FF maps with a low NRMSE error with a 20-fold reduction of the processing time. The time required by the trained network for $R2^*$/FF maps generation (about 20 ms) is compatible with the use into interactive software, where the used can explore different pancreatic regions with a near real-time feedback. The develop U-Net also mimics the ability of the GT fitting algorithm to minimize the estimation error in the $R2^*$/FF range of clinical interest. The high SSIM value (0.95) assessed in the $R2^*$ map comparison confirms the quality of $R2^*$ map generation by the developed CNN, while a lower SSIM value was found for FF maps.

The clinical $R2^*$ value estimation within the three main pancreatic regions revealed no significant bias between GT and CCN (Figure 4a). BA limits are comparable with inter-operator reproducibility in the clinical setting [17]. Higher estimation errors were found in patients with a severe iron overload where, due to the fast signal decay, the signal could be masked by MR noise at longer TEs. The assessed ICC value (0.9938) is comparable with the one obtained from a multicentre comparison (0.995) [6]. Hence, GT and CNN approaches appear to be interchangeable in the clinical setting for the $R2^*$ value evaluation. As far as the FF values evaluation is concerned, the concordance between GT and CNN approaches was lower, with a significant difference between the measurements. CNN's approach slightly underestimated the FF value. Although the GT technique is well validated for contemporary $R2^*$ and FF evaluation in the liver [45,46], few reports are available for the pancreas [20], so a comparison of the obtained results with the reproducibility in the clinical setting is difficult. FF value estimation in the pancreas was reported to be challenging, with an estimated CoV among observers of about 11% [47], similar to the one assessed in the study.

Some limitations should be recognized in the present study. The study was limited to a single U-Net architecture and a single MR protocol. Training and testing were performed on a high-quality, homogeneous data set obtained from a national network [6,34]. Although the MR sequence design employed in the study is of common use in the clinical setting, an external verification using datasets from other clinical centers could confirm the clinical generalization. It is expected that transfer learning could be used to minimize future CNN training for different acquisition sequences.

In conclusion, deep learning utilizing a U-Net is a feasible method for MR fat-corrected relaxometry. A trained U-Net can be efficiently used for MR fat-corrected relaxometry, providing results comparable to conventional model-based methods.

**Author Contributions:** Conceptualization, M.F.S. and A.M.; methodology, M.F.S.; software, S.J.; validation, S.J., A.M. and V.P.; data curation, L.P. and A.M.; data collection, T.C., F.M., P.P.B. and M.A.; writing—original draft preparation, M.F.S. and S.J.; writing—review and editing, V.P.; supervision, F.C.; project administration, F.C.; funding acquisition, F.C. All authors have read and agreed to the published version of the manuscript.

**Funding:** The MIOT project received "no-profit support" from industrial sponsorships (Chiesi Farmaceutici S.p.A.). The E-MIOT project receives "no-profit support" from industrial sponsorships (Chiesi Farmaceutici S.p.A. and Bayer).

**Data Availability Statement:** The image data are not publicly available due to ethical restrictions. The developed code is publicly available at https://github.com/vincenzopositano/Hippo-Net-Pancreas (accessed on 16 June 2022).

**Acknowledgments:** We thank Diego Hernando for allowing us to use his code to generate $R2^*$/FF maps. We also want to thank all the colleagues involved in the MIOT and EMIOT projects (https://emiot.ftgm.it, accessed on 16 June 2022).

**Conflicts of Interest:** The authors declare no conflict of interest. The funders had no role in the design of the study; in the collection, analyses, or interpretation of data; in the writing of the manuscript; or in the decision to publish the results.

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
