# Peer review of "Fat-Corrected Pancreatic R2* Relaxometry from Multi-Echo Gradient-Recalled Echo Sequence Using Convolutional Neural Network"

_electronics, doi:10.3390/electronics11182829_

Round 1

Reviewer 1 Report

This paper presents a U-Net architecture having as input mGRE images and returning the corresponding R2* and FF maps. The feasibility and performance of the proposed approach were assessed on clinical mGRE images acquired to estimate the iron overload level in the pancreas. The topic is interesting and matches well for Electronics journal. The paper contains a review of related works. I consider this paper to be an informative study and can be accepted for publication after only minor revision.

Specific comments:

1. The font of the text in figure 2 should be increased.

2. The font of the letters "a, b, c, d" in the description of Figure 3 should be reduced.

Author Response

This paper presents a U-Net architecture having as input mGRE images and returning the corresponding R2* and FF maps. The feasibility and performance of the proposed approach were assessed on clinical mGRE images acquired to estimate the iron overload level in the pancreas. The topic is interesting and matches well for Electronics journal. The paper contains a review of related works. I consider this paper to be an informative study and can be accepted for publication after only minor revision.

We would like to thank the reviewer for the appreciation of our work.

Specific comments:

  1. The font of the text in figure 2 should be increased.
  2. The font of the letters "a, b, c, d" in the description of Figure 3 should be reduced

Figures 2 and 3 were modified according to the reviewer’s suggestions.

Reviewer 2 Report

Dear Authors

I have read Fat-corrected pancreatic R2* relaxometry from multi‐echo  gradient‐recalled echo sequence using convolutional neural network  send to Electronics . The article is interesting. The subject is very interesting. I have some suggestions to consider before final publication. In the introduction, the Authors did not specify the purpose of the research. What is the utilitarian aspect of research. 

Author Response

Dear Authors

I have read Fat-corrected pancreatic R2* relaxometry from multi‐echo  gradient‐recalled echo sequence using convolutional neural network  send to Electronics . The article is interesting. The subject is very interesting. I have some suggestions to consider before final publication. In the introduction, the Authors did not specify the purpose of the research. What is the utilitarian aspect of research.

We would like to thank the reviewer for the appreciation of our work. We have modified the introduction of the paper to highlight the scope of the research. In particular, the final sentence in the introduction was modified as:

“The objective of the present study is to develop a computationally efficient method, based on a deep learning approach, to obtain precise and fast generation of fat-corrected pancreatic R2* maps. The feasibility and performance of the proposed method were assessed on clinical mGRE images acquired to estimate the iron overload level in the pancreas.”

Reviewer 3 Report

This paper studied the fat-corrected pancreatic R2* relaxometry from multi-echo gradient-recalled echo sequence using convolutional neural network, and obtained some corresponding simulation results. The topic is of some interest. However the motivation of the work is not clear, and the methods used in this paper is not new.

Author Response

This paper studied the fat-corrected pancreatic R2* relaxometry from multi-echo gradient-recalled echo sequence using convolutional neural network, and obtained some corresponding simulation results. The topic is of some interest. However the motivation of the work is not clear, and the methods used in this paper is not new.

We would tank the reviewer for the useful comments.

We have modified the introduction of the paper to highlight the scope of the research. The main motivation was to reduce the processing time needed for the generation of fat-corrected R2* maps, to spread their use in clinical practice. The second objective was to demonstrate the feasibility of fat-corrected R2* relaxometry in the pancreas, as previous studies were mainly performed in the liver and other organs.

We understand that the CNN architecture used in the study is not new. The introduction and the conclusions sections were modified in accordance with the reviewer's comments. In particular:

We add the following sentence to the introduction:

“Moreover, in our knowledge, the use of CNNs in the pancreatic region was not yet explored."   

The final sentence of the introduction was modified as:

“The objective of the present study is to develop a computationally efficient method, based on a deep learning approach, to obtain precise and fast generation of fat-corrected pancreatic R2* maps. The feasibility and performance of the proposed method were assessed on clinical mGRE images acquired to estimate the iron overload level in the pancreas.”

We add the following sentence to conclusions:

“U-Net architectures similar to the one presented in our study were proposed to address the water/fat separation problem [25–28], with different anatomical targets. The pancreas represents a complex structure affected by high chemical shift effect due to infiltrating fat, hence the development of an efficient R2* map generation could be challenging.”       

Round 2

Reviewer 3 Report

The article has hardly been revised according to the comments. The problem still exists. I am afraid I can not recommend. 
